# Coronatine Enhances Chilling Tolerance of Tomato Plants by Inducing Chilling-Related Epigenetic Adaptations and Transcriptional Reprogramming

**DOI:** 10.3390/ijms231710049

**Published:** 2022-09-02

**Authors:** Ziyan Liu, Zhuoyang Li, Shifeng Wu, Chunxin Yu, Xi Wang, Ye Wang, Zhen Peng, Yuerong Gao, Runzhi Li, Yuanyue Shen, Liusheng Duan

**Affiliations:** 1Beijing Key Laboratory for Agricultural Application and New Technique, College of Plant Science and Technology, Beijing University of Agriculture, Beijing 102206, China; 2State Key Laboratory of Plant Physiology and Biochemistry, College of Biological Sciences, China Agricultural University, Beijing 100193, China; 3State Key Laboratory of Plant Physiology and Biochemistry, College of Agronomy and Biotechnology, China Agricultural University, Beijing 100193, China

**Keywords:** chilling tolerance, transcriptional reprogramming, epigenetic adaptations, coronatine, tomato, ChIP-seq

## Abstract

Low temperature is an important environmental factor limiting the widespread planting of tropical and subtropical crops. The application of plant regulator coronatine, which is an analog of Jasmonic acid (JA), is an effective approach to enhancing crop’s resistance to chilling stress and other abiotic stresses. However, the function and mechanism of coronatine in promoting chilling resistance of tomato is unknown. In this study, coronatine treatment was demonstrated to significantly increase tomato chilling tolerance. Coronatine increases H3K4me3 modifications to make greater chromatin accessibility in multiple chilling-activated genes. Corresponding to that, the expression of *CBFs*, other chilling-responsive transcription factor (TF) genes, and JA-responsive genes is significantly induced by coronatine to trigger an extensive transcriptional reprogramming, thus resulting in a comprehensive chilling adaptation. These results indicate that coronatine enhances the chilling tolerance of tomato plants by inducing epigenetic adaptations and transcriptional reprogramming.

## 1. Introduction

Low temperature is a major abiotic stress that adversely impacts the growth, yield, and quality of crops. Many tropical and subtropical crops, such as tomato, rice, and maize, are extremely sensitive to low temperatures of 0–12 °C and unable to tolerate freezing [1,2]. In contrast, temperate plants such as wheat and *Arabidopsis* acquire freezing tolerance by exposure to low temperatures above freezing, a process known as cold acclimation [1,2]. Three cold-induced CBF (C-repeat-Binding Factor)/DREB1 (Dehydration-Responsive Element-Binding protein 1) transcription factors (TFs) play essential roles in cold acclimation [3,4,5,6,7]. These CBF/DREB1 TFs directly bind to the promoters of cold-regulated (*COR*) genes to induce their expression, thus enhancing the freezing tolerance of *Arabidopsis* [1,5,8,9]. Hundreds of *COR* genes have been identified as CBF regulons through *cbf1,2,3* (*cbfs*) triple mutants [6,7]. On the other hand, cold-induced CBF regulon genes are co-regulated by other first-wave TFs, such as WRKY33, MYB44, MYB73, ZF (ZINC FINGER), CZF1 (Cold-induced ZINC FINGER proteins 1), CZF2, ZAT10, ERF5 (Ethylene Response Factor 5), and so on [10]. Several first-wave TF genes, including *CZF2*, *ZAT10*, *MYB44*, and *MYB73*, are also the direct targets of the CBFs. Therefore, the CBFs function as key regulators, interconnecting multiple first-wave TFs to amplify cold responses [3].

CBF-like proteins are conserved and functional in a wide range of plants, such as barley [11], rice [12], and tomato [2,12]. There are three *CBF* homologs (*Lycopersicon esculentum CBF1-CBF3*) in tomato. Among them, the *LeCBF1* gene is significantly induced by 4 °C treatment and increases the freezing tolerance of *LeCBF1* transgenic *Arabidopsis* plants. However, the CBF regulons in tomato are obviously smaller and less diverse in function [2]. Through 7-day of exposure to 10 °C, tomato plants obtain a chilling acclimation response, including extensive transcriptomic and metabolic adjustments, thus enhancing chilling tolerance. More importantly, chilling and cold acclimation are regulated by similar transcription factors and hormones [13]. Besides that, some transcription factors in tomato, such as the basic helix–loop–helix (bHLH) transcription factor SlICE1 (Inducer of CBF Expression 1), SlMYC2, and SlNAM1 (NAC-type transcription factor), also enhance chilling tolerance of tomato or tobacco [14,15,16]. However, the *LeCBF1* and *SlICE1* transgenic tomato plants are stunted in growth, thereby the application of plant regulators may be another effective way to enhance the chilling tolerance of tomato.

When exposed to low temperatures, plants synthesize a large number of protective substances to enhance cold tolerance. The increase in soluble sugars, such as sucrose, raffinose, and trehalose, enhances plant cold tolerance by maintaining cell osmotic potential and cell membrane stability [17,18]. Exogenously application of oligosaccharides, including chitosan oligosaccharide (CSOS), cello-oligosaccharide (COS), and xylooligosaccharide (XOS), effectively increases the content of soluble sugars and plant chilling tolerance [19]. Under chilling stress, plants accumulate polyamines to increase resistance to chilling [20,21]. In addition, plants enhance the activity of antioxidant enzymes to prevent cold-induced oxidative stress and photosynthetic damage [22,23].

Epigenetic modifications create various chromatin states for the expression of stress-responsive genes that is considerable for abiotic stress resistance [24,25]. Through chromatin immunoprecipitation and deep sequencing (ChIP-seq), several types of histone modifications have been shown to be associated with gene expression in many organisms [26,27,28,29]. Osmotic stress slightly increases H3K4me3 modifications at the genomic level in *Arabidopsis* [30]. Dehydration- and ABA-induced genes present significantly broader H3K4me3 distribution profiles before and after the stress [31]. In addition, H3K27me3 modifications in two *COR* genes, *COR15A* (*Cold Regulated 15A*) and *GOLS3* (*Galactinol Synthase 3*), decrease gradually during exposure to low temperature and function as a memory marker for recent transcriptional activity [32,33]. Although some histone modifications have been shown to involve in cold responses, little is known about genome-wide dynamic changes of histone modifications induced by chilling stress.

Coronatine is a non-host-specific phytotoxin produced by *Pseudomonas syringae*, which helps the pathogen obtain higher pathogenicity [34,35,36]. This phytotoxin mimics the structure and function of (3R,7S)-jasmonoyl-L-isoleucine (JA-Ile), which is the bioactive form of plant hormone JA [37,38]. A high concentration of coronatine results in adverse impacts for several plants, including disease symptoms and leaf chlorosis [39,40], while the low concentration of coronatine increases crop resistance to multiple abiotic stresses, such as chilling stress in cucumber (*Cucumis sativus* L.), salinity stress in cotton (*Gossypium hirsutum*), and drought stress in soybean (*Glycine max*) [41,42,43]. Therefore, coronatine is an effective and environmentally friendly plant regulator in promoting stress tolerance of crops. However, the exact molecular mechanism of its action remains unclear. Based on previous studies, it was hypothesized that coronatine could also enhance the chilling tolerance of tomato, and this hypothesis needed experimental verification.

In this study, the tomato resistance to chilling stress is significantly improved by coronatine treatment. Transcriptomic analysis and genome-wide ChIP-seq analysis by H3K4me3 showed that coronatine enhances H3K4me3 modifications in multiple chilling responsive genes, including *CBFs*, other first-wave TF genes, and JA-responsive genes, to activate their expression and amplify chilling responses, thus enhancing chilling tolerance of tomato.

## 2. Results

### 2.1. Coronatine Increases the Chilling Tolerance of Tomato Plants Dependent on CBF Pathway

To test the effect of coronatine on the chilling tolerance of tomato, 0–4 nM coronatine was sprayed onto 17-day-old tomato leaves for 24 h. After 2-day chilling exposure, coronatine significantly enhanced the chilling tolerance of tomato seedlings compared to the control (Figure 1a). Ion leakage and relative injured area indicate injury caused by chilling stress. Correspondingly, coronatine treatment significantly decreased the ion leakage and relative injured area of tomato leaves (Figure 1b,c). Furthermore, statistical analysis showed that the most efficient concentration of coronatine was 1 nM (Figure 1a–c). These results indicate that coronatine pretreatment is an effective way to elevate the chilling tolerance of tomato plants.

To further explore whether coronatine regulates tomato chilling tolerance dependent on the CBF pathway, the expression of three *SlCBF**s* was examined in tomato seedlings, which were pretreated with or without 1 nM coronatine for 24 h and then exposed to 4 °C for indicated times. RT-qPCR analysis showed that the expression levels of *SlCBF1* and *SlCBF2* were significantly induced by coronatine treatment (Figure 2a). Moreover, chilling-induction of *SlCBF1* was also significantly increased in coronatine treatment samples (Figure 2a). Because coronatine is the analog of JA-Ile, JA responses in these tomato seedlings were also determined. Expression levels of *Protein Inhibitor II* (*PI-II*) and *Threonine Deaminase* (*TD*) [44] were significantly induced by coronatine (Figure 2b), indicating that the coronatine treatment was efficient. These results suggest that coronatine pretreatment positively regulates the CBF pathway to improve the chilling tolerance of tomato.

### 2.2. Coronatine Promotes the Expression of a Wide Range of Chilling-Responsive Genes through Transcriptional Reprogramming

To identify chilling and coronatine responsive genes in tomato, RNA-seq analysis was performed to study the transcription level of genes in wild-type (AC) tomato seedlings, which were pretreated with or without 1 nM coronatine for 24 h and then exposed to 4 °C for 0 or 12 h. Through Illumina sequencer “Novaseq 6000”, 20 million reads per sample were obtained, and more than 90% of these reads could be mapped to the tomato genome (Slycopersicum_691_ITAG4.0) (https://solgenomics.net). A total of 18,745 expressed genes were identified, which were expressed in at least one of the 12 samples (Appendix A). Principal component analysis (PCA) showed that the 12 samples were grouped into four categories (Appendix A), indicating that the samples of different treatments can be separated well.

Differentially expressed genes (DEG) in 4 °C or coronatine treatment samples vs. control samples were identified based on the filter criteria Log2 (|Fold Change|) ≥ 1, *p* < 0.05. Compared to control, 1119 upregulated genes and 1180 downregulated genes were identified in 4 °C treatment samples (Figure 3a and Appendix A). Gene Ontology (GO) enrichment analysis showed that the chilling-regulated genes were significantly enriched in the biological process “microtubule-based process” and “cell death”; and the molecular function GO terms “protein serine/threonine kinase activity”, “molecular transducer activity”, “signaling receptor activity”, “transcription factor activity”, and so on (Figure 3b and Appendix A).

Meanwhile, a total of 596 coronatine-induced genes and 88 coronatine-suppressed genes were identified in 1 nM coronatine treatment samples (Figure 3c), indicating that coronatine mainly functions in gene activation. Further GO enrichment analysis showed that coronatine-responsive genes were obviously enriched in biological process GO terms “responses to wounding” and “microtubule-based process”; and molecular function categories of transcription factor activity, identical protein binding, lipase activity, transmembrane signaling receptor activity, and so on (Figure 3d, Appendix A). In addition, 1055 upregulated genes and 1180 downregulated genes were identified in coronatine and 4 °C co-treatment samples when compared to samples only treated with 4 °C (Appendix A).

To further test the mechanism of coronatine enhancing chilling tolerance of tomato, the relationship between chilling and coronatine responsive genes was analyzed. A total of 412 co-upregulated genes and 71 co-downregulated genes were identified (Figure 4a and Appendix A), revealing that the chilling and coronatine co-responsive genes are mainly activated genes. Among these 412 co-upregulated genes, 273 chilling-activated genes could be upregulated by coronatine treatment alone, and the expression of other 139 chilling-activated genes could be enhanced by coronatine and 4 °C co-treatment (Figure 4a and Appendix A). Therefore, 412 genes were defined as coronatine-induced chilling responsive genes. GO enrichment analysis showed that 412 coronatine-induced chilling responsive genes were enriched in the biological process “microtubule-based process”; and the molecular function GO terms “signaling receptor activity”, “molecular transducer activity”, “protein serine/threonine kinase activity”, “calmodulin binding”, “transcription factor activity”, and “dioxygenase activity” (Figure 4b, Appendix A). Moreover, KEGG pathway analysis revealed that these 412 genes were mainly enriched in “signal transduction”, “environmental information processing”, “plant hormone signal transduction”, and “biosynthesis of other secondary metabolites” (Appendix A). These results suggest that the coronatine-induced chilling responsive genes are mainly involved in signal transduction and environmental cues, which are very important for enhancing the chilling tolerance of tomato seedlings.

### 2.3. Chilling-Induced CBFs, Other First-Wave TF Genes, and JA-Responsive Genes Play Essential Roles in Coronatine-Mediated Chilling Tolerance

Multiple cold-induced TF genes in first-wave of cold response, containing *WRKY33*, *COLD INDUCED ZINC FINGER PROTEIN 2* (*CZF2*), *MYB44*, *MYB73,* and *NAC domain protein*, are directly regulated by CBFs and coordinate with CBFs to amplify cold response [3,6,10]. Same as that, the expression levels of more than 20 TF genes, such as *WRKYs*, *ZINC FINGER proteins* (*ZF proteins*), *AUXIN RESPONSE FACTORs* (*ARFs*), and *NAC domain proteins*, were significantly upregulated by chilling, coronatine, and co-treatment (Figure 5a, Appendix A). Further RT-qPCR tests confirmed that these *WRKYs* and *ZF* genes were distinctly upregulated by chilling, coronatine, and co-treatment (Figure 5b), indicating that pretreatment of coronatine triggers the expression of multiple chilling-activated TF genes, thus coordinating with CBFs to amplify chilling-response.

Furthermore, according to previous studies [32], about 18% (73 genes) of the coronatine-induced chilling responsive genes were also JA-responsive genes (Figure 6a,b, Appendix A). Among them, JA biosynthetic gene *TomLOXD* (*LIPOXYGENASE D*) and protein kinase *MAPK5* (Figure 6b) are well-characterized early JA-responsive genes [45,46]. These results suggest that coronatine-induced JA-responsive genes also play important roles in promoting tomato resistance to chilling stress.

### 2.4. Identification of the Genome-Wide H3K4me3 Profiles in Tomato

To dissect the genome-wide histone modifications during coronatine-triggered transcriptional activation, ChIP-seq assays were performed using tomato seedlings treated with or without coronatine, and two independent biological replicates were performed. Same as previous studies on H3K4me3 profiles in animals and plants [26,27], the H3K4me3 signals mostly cover a region around the transcription start sites (TSS) (Figure 7a). A total of 25,048, 24,451, 22,297, and 26,599 H3K4me3 peaks were identified in control and coronatine treatment samples of two independent biological replicates through MACS2 software (Figure 7b, Appendix A) [47]. Next, the H3K4me3 peaks were annotated using ChIPseeker software [48] and found that about 85% of H3K4me3 signals were enriched in genic regions (from 1 kb upstream of TSS to 300 bp downstream of TES) (Figure 7c). About half of tomato genes (~18,000 genes) were associated with H3K4me3 modifications both in control and coronatine treatment samples of two independent biological replicates (Figure 7d and Appendix A). These results reveal that H3K4me3 modifications mainly present in coding regions and may play important roles in the transcriptional regulation of tomato.

### 2.5. Coronatine-Enhanced H3K4me3 Modifications Are Associated with Transcriptional Reprogramming

To explore the function of H3K4me3 modifications in coronatine-mediated chilling tolerance, the H3K4me3 peaks in coronatine treatment samples were compared with that in untreated control samples. A total of 14,209 and 9215 coronatine-enhanced H3K4me3 peaks were identified in two independent biological replicates, and these peaks mostly cover a region around the TSS (Figure 8a and Appendix A). Through ChIPseeker software, 5804 overlap genes in which the H3K4me3 modifications were enhanced by coronatine treatment were identified (Figure 8b and Appendix A). The H3K4me3 peaks in coronatine-induced chilling responsive gene regions were further analyzed. Venn diagram showed that the H3K4me3 peaks in 68 coronatine-induced chilling responsive genes were enhanced by coronatine (Figure 8c and Appendix A). GO enrichment analysis revealed that the 68 genes were enriched in the molecular function GO terms “transcription factor activity” (Figure 8d and Appendix A). The H3K4me3 modifications on master TF genes *SlCBF1* and *SlCBF2* and other chilling-induced TF genes were significantly increased by coronatine treatment (Figure 9a,b and Appendix A). In addition, coronatine also enhanced the H3K4me3 modifications on JA-responsive genes such as *TomLoxD* and *MAPK5* (Figure 9b and Appendix A). These results indicate that coronatine increases H3K4me3 modifications to make greater chromatin accessibility in multiple chilling-activated genes, thus triggering an extensive transcriptional reprogramming to enhance the chilling tolerance of tomato.

## 3. Discussion

### 3.1. Coronatine Is Involved in JA- and Oxidative Reactions-Mediated Chilling Tolerance

Numerous studies have shown that the plant hormone JA positively regulates cold response in multiple plants. JA promotes the degradation of JASMONATE ZIM-DOMAIN (JAZ) proteins, which physically interact with ICE1 and repress the transcriptional function of ICE1, thereby enhancing CBF-dependent plant tolerance to freezing stress [49]. JA reduces the chilling injury of several tropical and subtropical fruits, such as mango, banana, and tomato fruits [50,51,52,53]. MeJA treatment significantly induces the expression of many cold-responsive genes under cold storage to reduce the damage to plants caused by cold [52]. In tomato, JA coordinates with melatonin and forms a positive feedback loop to amplify the cold responses [54]. Coronatine is an analog of JA-Ile and regulates amounts of JA-responsive genes in tomato leaves [37,39]. In this study, 73 JA-responsive genes were identified in the coronatine-induced chilling responsive genes, including JA biosynthetic gene *TomLoxD*, *MAPK5*, and the MYC2 repressor *JAZ1* (Figure 6b, Appendix A). These genes are not only well-characterized early JA-responsive genes [45,46] but also *COR* genes [13]. Consistent with that, H3K4me3 modifications on *TomLoxD* and *MAPK5* were distinctly enhanced by coronatine (Figure 9b and Appendix A). Moreover, coronatine-induced chilling responsive TFs, such as WRKYs and ZF proteins, also played critical roles in JA signaling pathway (Figure 5a) [45,46]. These results reveal that the JA pathway is closely related to coronatine-mediated chilling tolerance.

Cold-induced elevated levels of reactive oxygen species (ROS) damage DNA, protein, and the structure of biofilm, thus impacting plant development and metabolism [55,56]. Several antioxidant enzymes, including ascorbate peroxidase (APX), superoxide dismutase (SOD), and reduced glutathione (GSH), are involved in reducing ROS production. In closely related species, the activity of these enzymes in chilling-tolerant species seems to be higher than that in chilling-sensitive species [22]. Overexpression of peroxidase genes enhances plant resistance to cold stress [57,58,59]. Consistent with that, the GO term “dioxygenase activity” was significantly enriched in coronatine-induced chilling responsive genes (Figure 4b). Among them, the expression of multiple peroxidase genes was significantly induced by coronatine pretreatment (Appendix A), which may help plants accumulate more peroxidases, thus reducing the damage of chilling-induced ROS bursts.

### 3.2. Coronatine Is Implicated in the Early Events of Chilling Response

Recently, protein kinases have been reported to participate in early events of cold signal transduction [60]. In *Arabidopsis*, OPEN STOMATA 1 (OST1) is activated by cold stress and phosphorylates ICE1 to inhibit the degradation of ICE1, thus positively regulating CBF expression and freezing tolerance [61]. COLD-RESPONSIVE PROTEIN KINASE 1 (CRPK1) phosphorylates 14-3-3 proteins in the cytoplasm, which triggers 14-3-3s to translocate into the nucleus to promote the degradation of CBFs [62]. Moreover, protein kinase MPK3/MPK6 negatively regulates ICE1 and freezing tolerance [63]. In tomato, protein kinases and phosphorylation-mediated signaling transduction play essential roles in the regulation of chilling tolerance [64]. A receptor-like protein kinase NECROTIC DWARF (NDW) was identified by map-based cloning and proved to positively regulate the chilling tolerance of tomato [65]. In addition, MULTIPLE RESISTANCE-ASSOCIATED KINASE 1 (MRK1) function as a novel positive regulator of multiple stresses, including chilling and heat stress [66]. In this study, coronatine-induced chilling responsive genes were enriched in the molecular function GO term “protein serine/threonine kinase activity” (Figure 4b). Among these protein kinases, several well-characterized stress-responsive genes such as *MAPK5*, *FLS2,* and *ATN1-LIKE PROTEIN* were identified (Appendix A), indicating that the coronatine is related to phosphorylation-mediated cold signaling transduction. Further investigation of the molecular mechanisms by which these kinases are activated by cold and coronatine will help us to understand early events of cold response in tomato.

Plant photosynthesis is sensitive to low temperatures [67]. The photosynthetic parameters include maximum quantum yield of photosystem II (Fv/Fm), performance index (PIABS), total chlorophylls (Chls), net photosynthetic rate (Pn), intercellular CO_2_ concentration (Ci), and water use efficiency (WUE) are significantly reduced by cold stress [23,68]. The decrease in these photosynthetic parameters is recovered under long-term cold treatment [23], which may be related to cold acclimation. Some of the *COR* genes encode chloroplast-associated proteins such as COR15a and COR15b, which protect chloroplast membrane and stromal proteins, thus reducing the inhibition of photosynthesis caused by cold stress [69,70]. In this study, coronatine-induced chilling responsive genes also contained chloroplast-associated genes, including *CYTOCHROME B5 REDUCTASE*, *LIPOXYGENASE 3*, and *ATP CARRIER PROTEIN 1*, which may function in mitigating the damage to the photosystem and the inhibition of photosynthesis under chilling stress.

Calcium signal and calcium channels/transporters function in cold sensing and early signaling transduction [60]. In rice, COLD1 interacts with the G-protein α subunit to activate the Ca^2+^ channel for sensing cold signal [71]. The Ca^2+^ transporters ANNEXIN1 (ANN1) and ANN4 were shown to involve in the cold-induced cytosolic Ca^2+^ influx and freezing tolerance in *Arabidopsis* [72]. In tomato, calcium-related kinases, such as CPK27 (Calcium-dependent Protein Kinases) and CaCIPK13 (Cold-induced CBL-Interacting Protein Kinase 13), also positively regulate cold response [73,74]. Consistent with that, the molecular function GO term “calcium binding” was also enriched in coronatine-induced chilling responsive genes in tomato (Figure 3b,d and Figure 4b). According to this, coronatine is speculated to trigger early events similar to chilling stress, thus resulting in high expression of hundreds of chilling-regulated genes.

### 3.3. Epigenetic Modifications Play Critical Roles in the Coronatine-Mediated Chilling Tolerance

In eukaryotes, epigenetic modifications produce various chromatin states for regulating stress-responsive gene expression, thus allowing plants to adapt to harsh and changeable environmental conditions [24,25]. Under chilling stress, the levels of histone modifications are altered, corresponding with gene expression adjustment [24,25,75]. A variety of histone modifications, H3K27me3, H3K9ac, and H4K5ac, were shown to be involved in chilling stress [76,77,78]. In potato (*Solanum tuberosum*), chilling stress enhances H3K4me3-H3K27me3 bivalent modifications and chromatin accessibility of active genes [79]. Indeed, H3K4me3 modifications in numerous coronatine-induced chilling responsive genes, including *SlCBFs* and other first-wave TF genes, kinases involved in early events of chilling response, and JA-responsive genes, were extensively induced by coronatine (Figure 8 and Appendix A). Therefore, coronatine pretreatment widely enhances the chromatin accessibility of chilling-activated genes to increase their expression, thus enhancing the chilling tolerance of tomato.

### 3.4. CBFs and Other First-Wave Transcription Factors Play Essential Roles in Coronatine-Mediated Chilling Tolerance Similar to “Chilling Acclimation”

Temperate plants such as wheat and *Arabidopsis* acquire freezing tolerance by cold acclimation, in which the CBF signaling pathway plays a critical role. The function and targets of CBFs have been extensively studied in *Arabidopsis* [3,4,5,6,7]. Hundreds of CBF regulons (cold-responsive genes regulated by CBFs) and 164 CBF directly targeted *COR* genes were identified [3]. Among them, first-wave transcription factors, such as R2R3-MYB, WRKY, ZINC FINGER proteins, and NAC domain proteins, function in the early events of cold response [3]. These first-wave transcription factors coordinate with CBFs to amplify the expression of *COR* genes. Different from temperate plants, tropical plants such as tomato and cassava acquire chilling tolerance by exposure to mild low temperature (10–14 °C), a process named chilling acclimation [13,80]. A study in tomato has shown that chilling and cold acclimation are regulated by similar transcription factors, such as CBFs, WRKYs, MYBs, bZIPs, ZINC FINGERs, and bHLHs [13]. Consist with that, over 20 transcription factors were identified in tomato chilling responsive genes (Appendix A) and were classified mainly into the WRKY, AP2/ERF, bHLH, bZIP, ZINC FINGER, MYB, ARF, and NAC families. A lot of these transcription factors were also enriched in coronatine-induced chilling responsive genes (Figure 4 and Figure 5a). These results demonstrate that CBFs and other first-wave transcription factors play essential roles in coronatine-mediated chilling tolerance, similar to “chilling acclimation”.

More importantly, H3K4me3 modifications on these transcription factors were significantly increased by coronatine. Because the H3K4me3 modifications are mainly associated with transcriptional activation, these results indicate that the coronatine pretreatment enhances the H3K4me3 modifications in chilling-induced TF genes to activate their expression, thus amplifying the chilling response in tomato.

## 4. Materials and Methods

### 4.1. Plant Material and Growth Conditions

Tomato (*Solanum lycopersicum*) cv Ailsa Craig (AC) plants were used for this research. For germination, tomato seeds were placed on madid filter paper for 3–4 days. Germinated seeds with uniform radical lengths were transferred into pots (7 cm × 7 cm × 8.5 cm, length × width × depth) and one tomato seedling per pot. The culture medium was a mixture of 66% soil (Pindstrup substrate, 10–30 mm, Denmark) and 34% vermiculite. Plants were grown in a greenhouse maintained at 25 °C and 60% relative humidity with daily cycles of 16 h of light (light intensity, 200 μmol photons m^−2^ s^−1^) and 8 h of dark.

### 4.2. Production and Preparation of Coronatine

Coronatine used in this work was produced by microbial fermentation and purified by the Centre for Crop Chemical Control, China Agricultural University. The fermentation broth of *Pseudomonas syringae* was extracted with organic solvent, further concentrated, and purified by column chromatography to obtain the crude product, and then recrystallized to obtain the pure product. The purity of coronatine was greater than 99%, measured by high-performance liquid chromatography (Milford, MA, USA). The coronatine product can be purchased from Chengdu NewSun Crop Science Co., Ltd. in China. In total, 958.2 μg coronatine was dissolved in 100 μL methanol and then diluted with water to 1 mM for storing. Before treatment, the mother solution of coronatine was diluted to a working concentration by water.

### 4.3. Plant Treatment

Coronatine treatments were performed as described in [42] with minor modifications. Seventeen-day-old tomato seedlings with two fully expanded leaves were sprayed with 0.25, 0.5, 1, 2, and 4 nM of coronatine, and the amount of liquid per plant was 2.5 mL. Tomato seedlings treated with water, which contained the same amount of methanol as the coronatine solution, were used as control. For chilling treatment, tomato seedlings pretreated with or without coronatine for 24 h were put into a growth chamber maintained at 4 °C and 60% relative humidity for 2 days. Then, the seedlings were transferred to a greenhouse maintained at 25 °C for additional 2 days before counting the injured area and ion leakage. Each treatment group contained 10 seedlings, and four independent biological replicates were performed. The relative injured area of first true leaf of every seedling was measured by ImageJ software and calculated by the following formula: Injured area (%) = injured area/full area × 100.

### 4.4. Ion Leakage Assays

Ion leakage assays were performed as described in [81] with minor modifications. After chilling treatment, the first true leaf of every seedling was collected into 50 mL tubes containing 40 mL of deionized water. Then the conductance of the water was detected as S_0_. The solution was shaken for 1 h at 25 °C and then measured as S_1_. The samples were boiled at 100 °C for 30 min, shaken at 25 °C for 1 h, and then detected as S_2_. Ion leakage was calculated by the ratio of (S_1_ − S_0_)/(S_2_ − S_0_).

### 4.5. RT-qPCR Assays

Through TRIzol reagent (Invitrogen, Thermo Fisher Scientific, USA), total RNA was extracted from the true leaves of tomato seedlings, which were sprayed with coronatine at 0 or 1 nM concentrations for 24 h and then exposed to 4 °C for 0 or 12 h. Then, the total RNA was reverse transcribed by M-MLV reverse transcriptase (Promega, Promega Corporation, USA). SYBR Green regent (Takara, TaKaRa Bio Group, Japan) was used for the quantitative real-time PCR assays. RT-qPCR analysis was performed as previously described in [82] with minor modifications. Briefly, relative transcript abundance was assessed using the comparative C_T_ method. The expression of tomato *ACTIN2* was used as a standard control. After calculation of ΔC_T_ (C_T, gene of interest_—C_T, actin2_), the relative expression level was calculated as 2^−^^ΔCT^. The 2^−^^ΔCT^ value of untreated WT plants was set to 1 for normalization [2^−^^ΔCT(WT(0 h))^/2^−^^ΔCT(WT(0 h))^ = 1]. The primers used for the RT-qPCR assays are listed in Appendix A.

### 4.6. RNA-Seq Analysis

For RNA-seq analysis, 17-day-old tomato seedlings were sprayed with coronatine at 0 or 1 nM concentrations for 24 h and then exposed to 4 °C for 0 or 12 h. Leaves of fifteen seedlings were harvested for each sample. In order to facilitate subsequent description, plants grown under normal conditions untreated with coronatine and chilling were named as control. Total RNA was extracted from these tomato leaves by TRIzol reagent (Invitrogen). Three independent biological replicates were performed for sequencing. Through Illunima NovaSeq 6000 at Beijing Berry Genomics, 20 million reads (150-bp paired-end) per sample were generated. Data analysis was performed as described in [3] with minor modifications. Briefly, FastQC (v.0.11.9) (http://www.bioinformatics.bbsrc.ac.uk/projects/fastqc) software was used for quality control of raw data (fastq format). Then reads were mapped to the tomato genome (Slycopersicum_691_ITAG4.0) by HISAT2 (v.2.2.0) with default parameters. The read counts were calculated by FeatureCounts (v.2.0.1) software. Then, R package DESeq2 (v.1.30.0) was used for calculating differentially expressed genes (DEGs) and principal component analysis (PCA). Transcripts Per Kilobase Million (TPM) values were used for DEGs analysis. Significant differential expression genes were identified by the filter criteria Log_2_ (|Fold Change|) ≥ 1, *p* < 0.05 (one-way ANOVA test). GO enrichment analysis of significant DEGs was performed by agriGO v2.0 [83]. TBtools software was used for performing the heatmap of DEGs (Chen et al., 2020).

### 4.7. ChIP-Seq Analysis

For ChIP-seq analysis, the tomato growth conditions were the same as RNA-seq samples, and two independent biological replicates were performed. Seventeen-day-old tomato seedlings sprayed with or without 1 nM coronatine for 24 h were harvested as ChIP samples and cross-linked with 1% formaldehyde for 10 min. The nuclei were isolated and resuspended in lysis buffer (5 mM Tris-HCl [pH 8.0], 10 mM EDTA [pH 8.0], 200 mM NaCl, 1% SDS, 1 mM DTT, 1 × inhibitor cocktail). The DNA fragment was sheared to 250 ~400 bp by sonication. Then, 50 μL of the sheared chromatin was used as the input control, and 10 μg anti-H3K4me3 antibody (Abcam, Cat# ab8580, RRID: AB_306649) was incubated with 30 μL aliquot of Dynabeads^TM^ Protein G (Invitrogen) for 8 h at 4 °C and then incubated with the remaining chromatin overnight at 4 °C. The immunoprecipitated complexes were eluted from the Dynabeads and incubation with 20 μL 5 M NaCl for 8 h at 65 °C. After that, DNA was recovered by MinElute PCR Purification Kit (QIAGEN). The libraries for sequencing were prepared by TransNGS^®^ Tn5 DNA Library Prep Kit for Illumina^®^ (KP121-11) and sequenced by Illunima NovaSeq 6000 at Beijing Berry Genomics. Then, 20 million reads (150-bp pair-end) per sample were generated by sequencing.

Data analysis was performed as described in [3] with minor modifications. Raw fastq data were processed by the Trim_Galore with Q20 (Phred score >20) for adaptor removal and quality control. After that, the cleaned reads were mapped to the tomato genome (Slycopersicum_691_ITAG4.0) through Bowtie2 (v 2.4.2). Using SAMtools (v1.4.1) with the MAQ20 (map quality > 20) filtration, unique aligned reads were processed for sequence alignment and mapping. Then, the peak calling was performed by MACS2 (v2.1.7) (bandwidth = 300; model fold = [5,50]; *q*-value cutoff = 0.01) [47]. The heatmaps were performed by deepTools2.0 (https://deeptools.readthedocs.io/en/develop/index.html).

### 4.8. Statistical Analysis

GraphPad Prism 8.0 software was used for Data sets analysis. Comparisons between different treatments were tested by the one-way ANOVA test. Comparisons between two samples were analyzed using the HSD Tukey test.

## 5. Conclusions

This work revealed that a low concentration of coronatine increases H3K4me3 modifications in multiple chilling-responsive genes, including *CBFs*, other first-wave TF genes, and JA-responsive genes, to make greater chromatin accessibility of these activated genes. Accordingly, the expression of *CBFs* and other first-wave TF genes is significantly induced by coronatine to trigger an extensive transcriptional reprogramming, including kinases and calcium binding genes related to early events of chilling responses, JA biosynthetic genes, and peroxidase genes, thereby obtaining a comprehensive chilling adaptation. Based on these mechanisms, coronatine pretreatment performs comparable function as “chilling acclimation”, thus promoting the chilling tolerance of tomato (Figure 10). This work demonstrated that coronatine can be used as an effective regulator to help tomato overcome harsh environmental conditions, and it will shed more light on the epigenetic regulation in plant chilling responses.

## Figures and Tables

**Figure 1 ijms-23-10049-f001:**
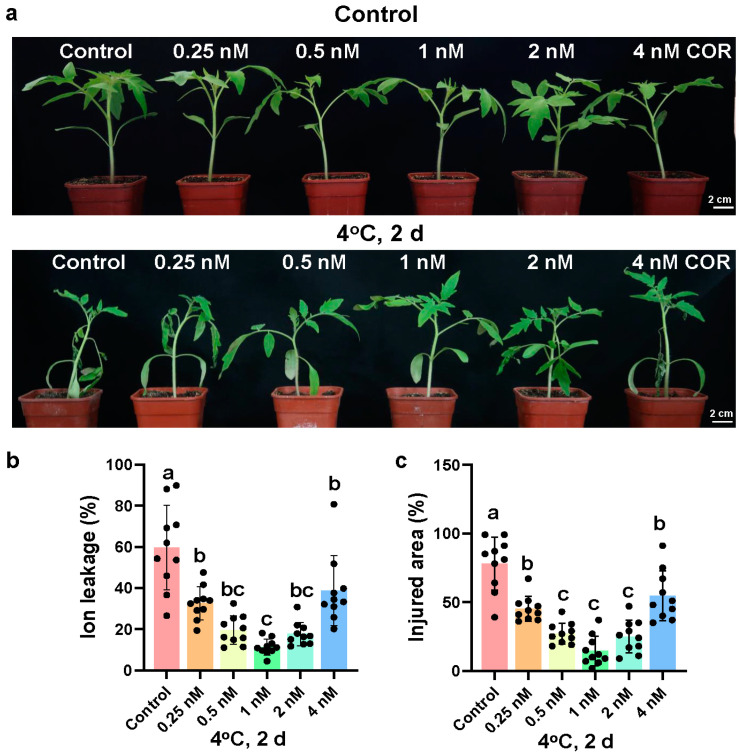
Coronatine enhances the chilling tolerance of tomato plants. (**a**–**c**) Chilling tolerance of tomato seedlings treated with or without coronatine. 17-day-old tomato seedlings were sprayed with coronatine at 0, 0.25, 0.5, 1, 2, or 4 nM concentrations for 24 h and then exposed to 4 °C for 2 days. After recovery at 25 °C for 2 days, representative images of tomato seedlings (**a**) were taken, ion leakage (**b**) and injured area (**c**) were measured. In (**a**), the picture above shows the representative tomato seedlings before exposure to 4 °C, and the picture below shows the representative tomato seedlings after exposure to 4 °C. In (**b**,**c**), bars represent means ± S.D. (*n* = 10), and the letters indicate significant differences by one-way ANOVA test (*p* < 0.05). Each small dot represents a single plant data.

**Figure 2 ijms-23-10049-f002:**
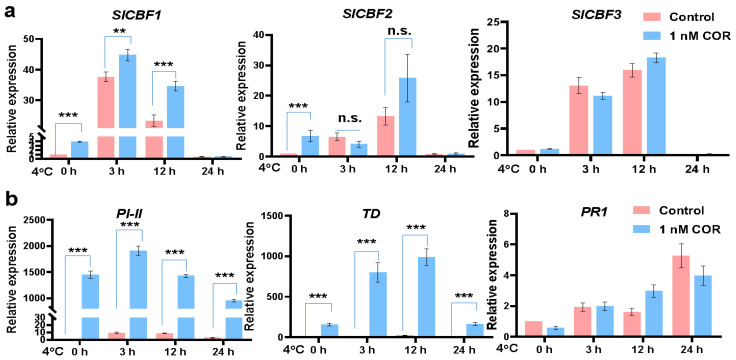
Coronatine induces the expression of *SlCBFs* and JA-responsive genes in tomato. (**a**) RT-qPCR showing the expression of *SlCBFs* in tomato seedlings. (**b**) RT-qPCR showing the expression of JA-responsive genes *PI-II*, *TD* and SA-responsive gene *PR1* in tomato seedlings. In (**a**,**b**), 17-day-old tomato seedlings were sprayed with or without 1 nM coronatine for 24 h, and then exposed to 4 °C for the indicated times. Total RNA was extracted and analyzed by RT-qPCR. Tomato *ACTIN2* was used as an internal control. The expression of indicated genes in untreated control plants was set at 1. Bars represent means ± S.D. (*n* = 3). The asterisks indicate significant differences by HSD Tukey test (** *p* < 0.01, *** *p* < 0.001, n.s. indicates no significant difference).

**Figure 3 ijms-23-10049-f003:**
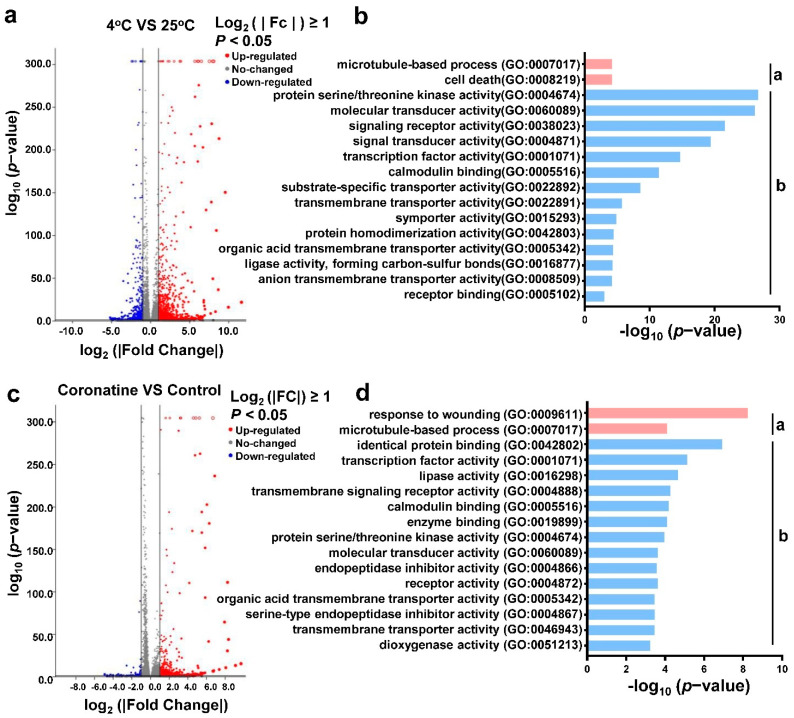
RNA-seq analysis showing differentially expressed genes (DEGs) in tomato chilling or coronatine responses. (**a**) Volcano plots showing the fold changes of gene expression levels (TPM, Transcripts Per Kilobase Million) in tomato seedlings exposure to 4 °C for 12 h compared with control. Genes which exceed the thresholds of Log_2_ (|Fold Change|) ≥ 1.0, *p*−value < 0.05 are shown as red dots (up-regulated) and blue dots (down-regulated). (**b**) GO classification of chilling-regulated genes. a, Biological Process; b, Molecular Function. (**c**) Volcano plots showing the fold changes of gene expression levels in tomato seedlings sprayed with coronatine 1 nM compared with control. (**d**) GO classification of coronatine-regulated genes. a, Biological Process; b, Molecular Function.

**Figure 4 ijms-23-10049-f004:**
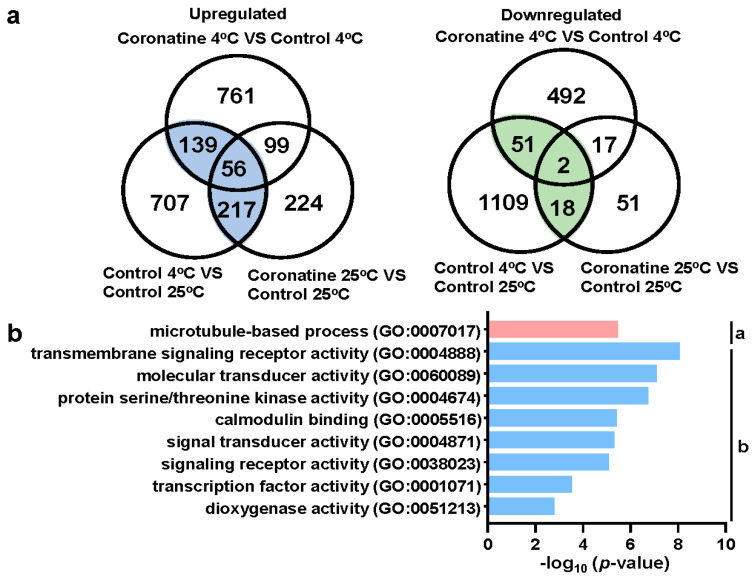
GO classification of coronatine-induced chilling responsive genes. (**a**) Venn diagram showing the 412 co-upregulated genes and the 71 co-downregulated genes in response to chilling and coronatine. (**b**) GO classification of 412 coronatine-induced chilling responsive genes. a, Biological Process; b, Molecular Function.

**Figure 5 ijms-23-10049-f005:**
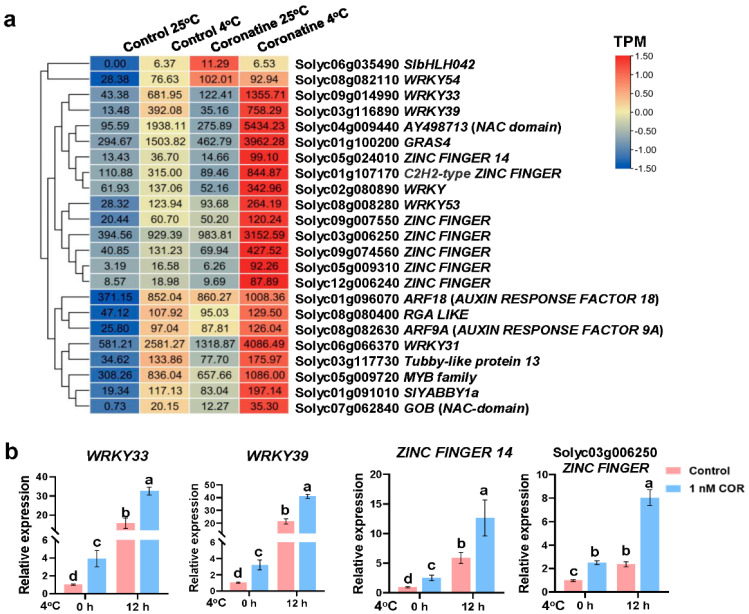
Transcription factor (TF) genes peaked out from coronatine-induced chilling responsive genes. (**a**) Heatmap showing the gene expression levels of TF genes peaked out from coronatine-induced chilling responsive genes with TPM (Transcripts Per Million). (**b**) Expression of *WRKY* and *ZINC FINGER* TF genes in tomato seedlings. 17-day-old tomato seedlings were sprayed with or without 1 nM coronatine for 24 h, and then exposed to 4 °C for 12 h. Total RNA was extracted and analyzed by RT-qPCR. Tomato *ACTIN2* was used as an internal control. The expression of indicated genes in untreated control plants was set at 1. Bars represent means ± S.D. (*n* = 3), and the letters indicate significant differences by one-way ANOVA test (*p* < 0.05).

**Figure 6 ijms-23-10049-f006:**
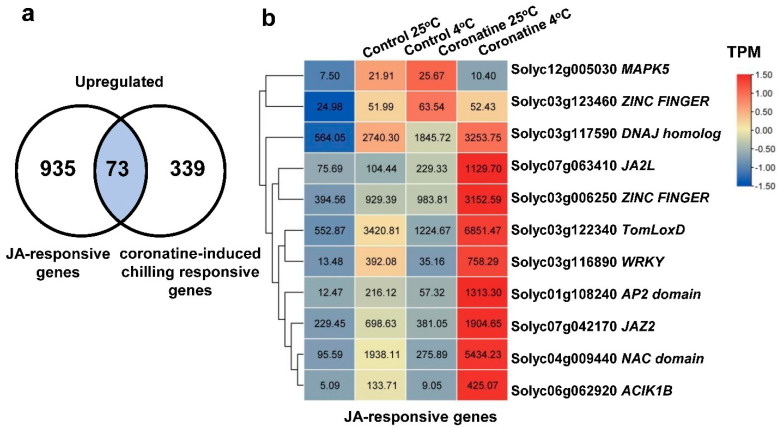
JA-responsive genes peaked out from coronatine-induced chilling responsive genes. (**a**) Venn diagram showing the 73 co-upregulated genes in response to chilling, coronatine and JA. (**b**) Heatmap showing the gene expression levels of JA-responsive genes peaked out from coronatine-induced chilling responsive genes with TPM.

**Figure 7 ijms-23-10049-f007:**
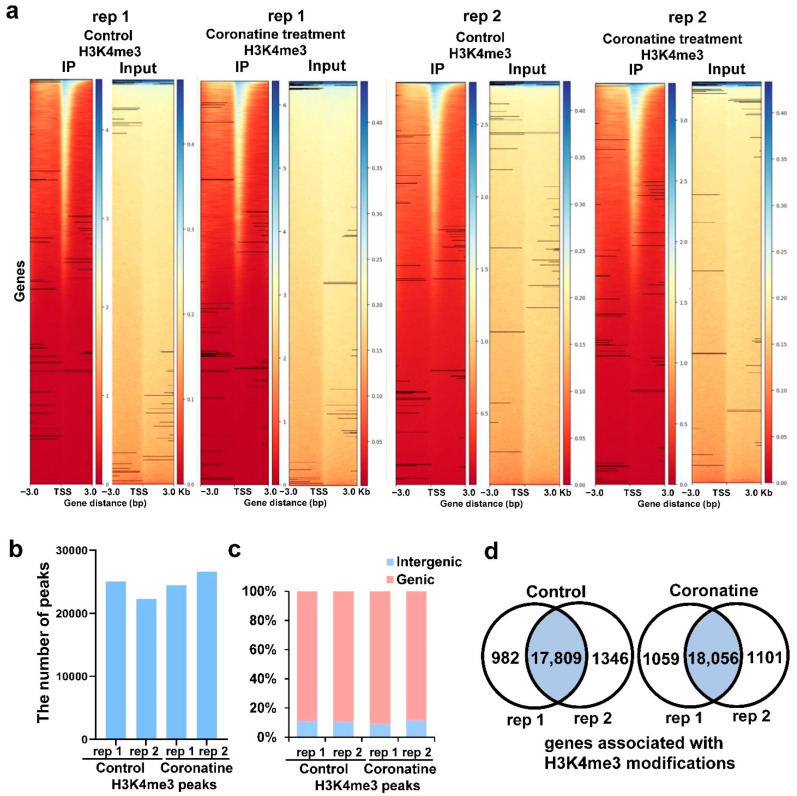
The characteristics of H3K4me3 peaks in the tomato genome from the ChIP-seq analysis. (**a**) Distribution of the H3K4me3 signals in tomato seedlings treated with or without coronatine. Each row represents a region of ±3 kb around the TSS and ranked by the H3K4me3 read counts in promoters. Two independent biological replicates are showed. (**b**) The number of H3K4me3 peaks in tomato seedlings treated with or without coronatine. (**c**) The proportion of H3K4me3 peaks in the genic or intergenic regions of the tomato genome. The genic regions include the gene body and the region 1 kb upstream of TSS and 300 bp downstream of TES. (**d**) Venn diagram showing the overlap genes associated with H3K4me3 modifications between two independent biological replicates.

**Figure 8 ijms-23-10049-f008:**
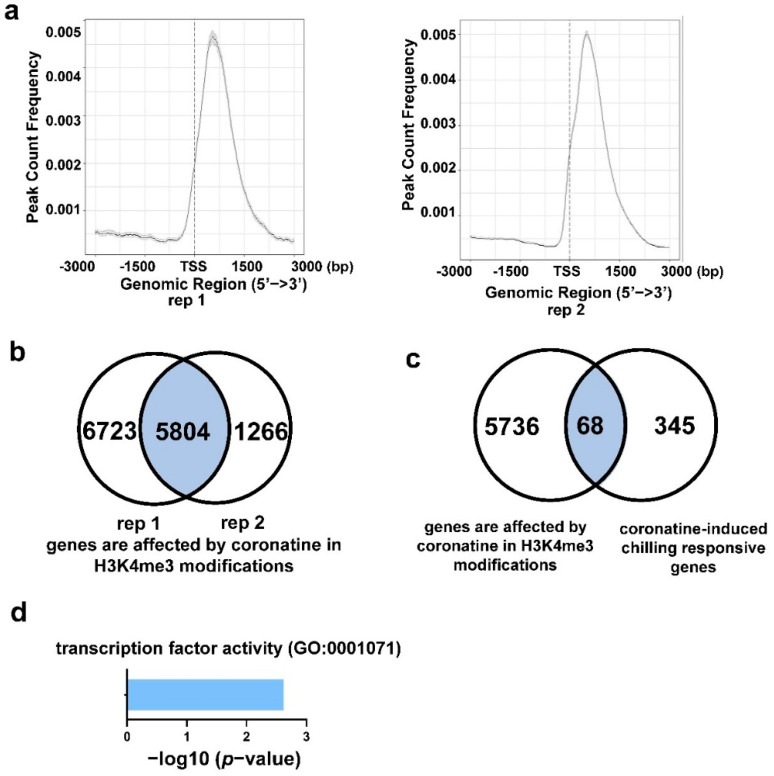
Coronatine-enhanced H3K4me3 modifications are associated with transcriptional regulation. (**a**) Average distance of the coronatine-enhanced H3K4me3 peaks flanking TSSs throughout the genome in two independent biological replicates. (**b**) Venn diagram showing the overlap genes that are affected by coronatine in H3K4me3 modifications between two independent biological replicates. (**c**) Venn diagram showing the overlap of genes are affected by coronatine in H3K4me3 modifications and chilling responsive genes upregulated by coronatine. (**d**) GO classification of 68 overlapped genes in (**c**).

**Figure 9 ijms-23-10049-f009:**
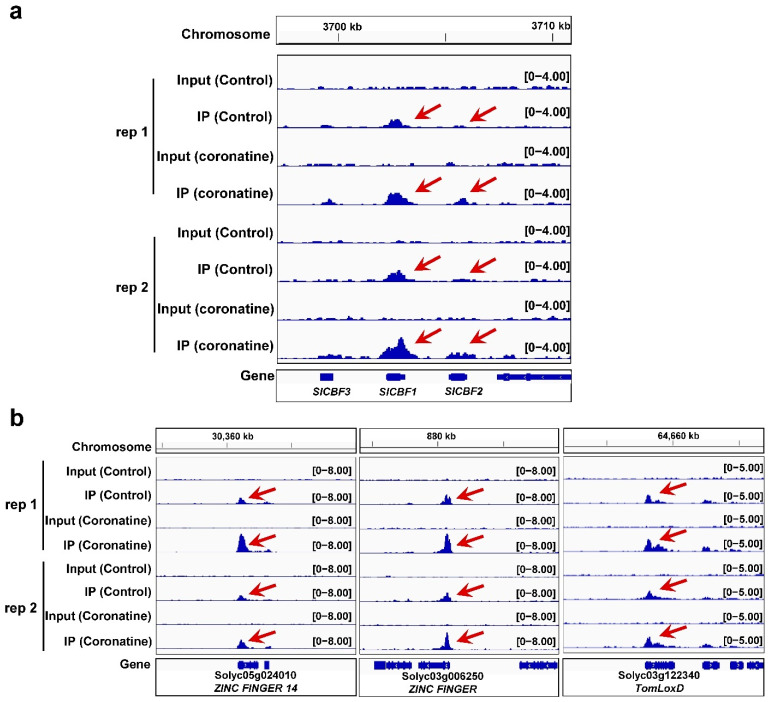
Genome browser view of the H3K4me3 modifications at the gene regions. (**a**) Genome browser view of the H3K4me3 RPM (reads per million reads) at the *SlCBFs* gene regions in two biological replicates. (**b**) Genome browser view of the H3K4me3 RPM at the typical *ZINC FINGERs* and JA-responsive gene *TomLoxD* regions in two biological replicates.

**Figure 10 ijms-23-10049-f010:**
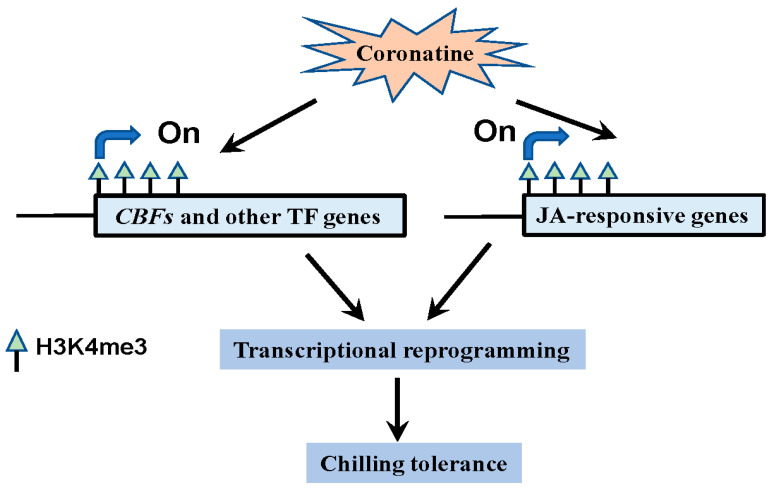
Coronatine enhances chilling tolerance of tomato plants by H3K4me3 modifications and transcriptional reprogramming. Coronatine increases H3K4me3 modifications to make greater chromatin accessibility in multiple chilling-activated genes. Corresponding to that, the expression of essential chilling-responsive genes, including *CBFs*, other first-wave TF genes and JA-responsive genes, is significantly induced by coronatine to trigger an extensive transcriptional reprogramming, thus enhancing chilling tolerance of tomato.

## Data Availability

The complete RNAseq data and ChIP-seq data from this publication have been submitted to the GEO database (http://www.ncbi.nlm.nih.gov/geo/) and assigned the identifier accession PRJNA823402. Sequence data used for this article can be found in the Sol Genomics Network Initiative or the National Center for Biotechnology Information under the following accession numbers: *SlCBF1*, Solyc03g026280; *SlCBF2*, Solyc03g124110; *SlCBF3*, Solyc03g026270; *PR1*, Solyc00g174340; *JA2L*, Solyc07g063410; *TomLoxD*, Solyc03g122340; *TD*, Solyc09g008670; *PI-II*, NP_001234627.1; *ACTIN2*, Solyc11g005330; *WRKY33*, Solyc09g014990; *WRKY39*, Solyc03g116890; *ZINC FINGER protein 14*, Solyc05g024010; and *WRKY54*, Solyc08g082110.

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
