# Peer review of "Coronatine Enhances Chilling Tolerance of Tomato Plants by Inducing Chilling-Related Epigenetic Adaptations and Transcriptional Reprogramming"

_ijms, 2022, doi:10.3390/ijms231710049_

Round 1

Reviewer 1 Report

This manuscript explored the effect of coronatine on low temperature tolerance in Tomato (Solanum lycopersicum) cv Ailsa Craig plants. It was found that the expression of CBFs and other first-wave TF genes was significantly induced by coronatine to trigger extensive transcriptional reprogramming. The authors also found that coronatine increased H3K4me3 modifications in multiple chilling-responsive genes. However, the following points should be considered to improve this work:

Abstract

1.     L16, Avoid first person narration, please check the full text.

2.     L16-17, Recommend deleting “In this study, we demonstrated that coronatine is an effective regulator in increasing tomato chilling tolerance.”. Please supplement the main results of the experiment.

Introduction

3.     L83, Please consider putting forward assumptions.

Results

4.     L127, Using an online analysis tool from AgriGO [38], Should be moved to the Materials and Methods section

5.     L179 and L190, Check citation format.

6.     L184, delete ”and histone H3 lysine 4 trimethylation (H3K4me3) antibodies,”.

Discussion

7.     The description of this part is confusing, please reorganize. Suggested order, 3.3-3.2-3.4-3.1

8.     L229, supplement examples of “chilling acclimation” and discuss it.

Materials and Methods

9.     L318, Why choose Tomato (Solanum lycopersicum) cv Ailsa Craig plant? Please explain

10.  L321, Soil type? Please explain.

11.  L323, the light intensity (200 μmol photons m-2 s-1) is too low.

12.  L323, Pot size, and seeding density? Please add.

13.  L325, Whether there are standard procedures for reference in the microbial fermentation process, please provide them.

14.  L341-342, Please specify how many biological replicates.

15.  L342, delete “and obtained similar results”.

16.  L351, 354, Please provide the brand, company and country of the main reagent.

17.  L381, It should be 4.7.

Author Response

This manuscript explored the effect of coronatine on low temperature tolerance in Tomato (Solanum lycopersicum) cv Ailsa Craig plants. It was found that the expression of CBFs and other first-wave TF genes was significantly induced by coronatine to trigger extensive transcriptional reprogramming. The authors also found that coronatine increased H3K4me3 modifications in multiple chilling-responsive genes. However, the following points should be considered to improve this work:

Abstract

  1. L16, Avoid first person narration, please check the full text.

Response: Thanks for the valuable suggestion and we have corrected it as suggested.

  1. L16-17, Recommend deleting “In this study, we demonstrated that coronatine is an effective regulator in increasing tomato chilling tolerance.”. Please supplement the main results of the experiment.

Response: Thanks for the valuable suggestion. We have changed the sentence as following: In this study, coronatine treatment was demonstrated to significantly increase tomato chilling tolerance. In Figure 1, we show the phenotype that the coronatine treatment significantly increase tomato chilling tolerance, the ion leakage and relative injured area were decreased by coronatine treatment, especially 1 nM coronatine treatment. And the RNA-seq and ChIP-seq data also support this point.

Introduction

  1. L83, Please consider putting forward assumptions.

Response: Thanks for the valuable suggestion. We have changed the sentence as following: Based on previous studies, it was hypothesized that coronatine could also enhance the chilling tolerance of tomato, and this hypothesis needed experimental verification.

Results

  1. L127, “Using an online analysis tool from AgriGO [38],” Should be moved to the Materials and Methods section

Response: Thanks for the valuable suggestion and we have moved it as suggested.

  1. L179 and L190, Check citation format.

Response: Thanks for the valuable suggestion and we have corrected it as suggested.

  1. L184, delete ”and histone H3 lysine 4 trimethylation (H3K4me3) antibodies,”.

Response: Thanks for the valuable suggestion and we have deleted it as suggested.

Discussion

  1. The description of this part is confusing, please reorganize. Suggested order, 3.3-3.2-3.4-3.1

Response: Thanks for the valuable suggestion and we have changed it as suggested.

  1. L229, supplement examples of “chilling acclimation” and discuss it.

Response: Thanks for the valuable suggestion. Besides the previous example of tomato chilling acclimation, we added the example of cassava. We have changed the sentence as following: Different from temperate plants, tropical plants like tomato and cassava acquire chilling tolerance by exposure to mild low temperature (10-14oC), a process named chilling acclimation (Barrero-Gil et al., 2016; Zeng et al., 2014). Study in tomato has shown that the chilling and cold acclimation are regulated by similar transcription factors, such as CBFs, WRKYs, MYBs, bZIPs, ZINC FINGERs and bHLHs (Barrero-Gil et al., 2016).

Reference:

Barrero-Gil, J., Huertas, R., Rambla, J.L., Granell, A., and Salinas, J. (2016). Tomato plants increase their tolerance to low temperature in a chilling acclimation process entailing comprehensive transcriptional and metabolic adjustments. Plant Cell Environ 39, 2303-2318.

Zeng, C., Chen, Z., Xia, J., Zhang, K., Chen, X., Zhou, Y., Bo, W., Song, S., Deng, D., Guo, X., et al. (2014). Chilling acclimation provides immunity to stress by altering regulatory networks and inducing genes with protective functions in cassava. BMC Plant Biol 14, 207.

Materials and Methods

  1. L318, Why choose Tomato (Solanum lycopersicum) cv Ailsa Craig plant? Please explain

Response: Thanks for the valuable suggestion. Because this tomato ecotype produces more fruit and lots of seeds, and easy to construct transgenic plants. It is the ecotype of tomato widely used in the research at present. At the same time, it is very sensitive to adverse environment, especially low temperature. It shows significant injury phenotype after 2 days of treatment at 4 ℃, which is a good model material for studying chilling response.

  1. L321, Soil type? Please explain.

Response: The soil was Pindstrup substrate (10 - 30 mm, Denmark), and we have added this information in our manuscript.

  1. L323, the light intensity (200 μmol photons m-2 s-1) is too low.

Response: Thanks for the valuable suggestion. The light intensity of 100 - 200 μmol photons m-2 s-1 is widely used in the study of the mechanism of tomato seedlings response to biotic stress and abiotic stress (Barrero-Gil et al., 2016; Du et al., 2017; Zhang et al., 2004). In practice, we find that tomato seedlings cultured in pot under the light intensity of 200 μmol photons m-2 s-1 have the best growth state, which is very suitable for both phenotype analysis and stress-induced gene expression analysis. The tomato seedlings grown in pot for 4 weeks were transplanted to the field, then it was more beneficial to increase the light intensity for the growth of tomato in the field.

Reference:

Barrero-Gil, J., Huertas, R., Rambla, J.L., Granell, A., and Salinas, J. (2016). Tomato plants increase their tolerance to low temperature in a chilling acclimation process entailing comprehensive transcriptional and metabolic adjustments. Plant Cell Environ 39, 2303-2318.

Du, M., Zhao, J., Tzeng, D.T.W., Liu, Y., Deng, L., Yang, T., Zhai, Q., Wu, F., Huang, Z., Zhou, M., et al. (2017). MYC2 orchestrates a hierarchical transcriptional cascade that regulates jasmonate-mediated plant immunity in tomato. Plant Cell 29, 1883-1906.

Zhang, X., Fowler, S.G., Cheng, H., Lou, Y., Rhee, S.Y., Stockinger, E.J., and Thomashow, M.F. (2004). Freezing-sensitive tomato has a functional CBF cold response pathway, but a CBF regulon that differs from that of freezing-tolerant Arabidopsis. Plant J 39, 905-919.

  1. L323, Pot size, and seeding density? Please add.

Response: Thanks for the valuable suggestion. We have added it as following: Germinated seeds with uniform radical lengths were transferred into pots (7 cm × 7 cm × 8.5 cm, length × width × depth) and one tomato seedling per pot.

  1. L325, Whether there are standard procedures for reference in the microbial fermentation process, please provide them.

Response: Thanks for the valuable suggestion and we have added the following information: The fermentation broth of Pseudomonas syringae was extracted with organic solvent, further concentrated and purified by column chromatography to obtain the crude product, and then recrystallized to obtain the pure product. The purity of coronatine was greater than 99%, measured by high performance liquid chromatography (Milford, MA, USA). The coronatine product can be purchased from Chengdu NewSun Crop Science Co., Ltd (China).

Since this coronatine product is already commercially produced and marketed as a pesticide, and this paper aims to study the function of coronatine but not production process, we cannot provide further details.

  1. L341-342, Please specify how many biological replicates.

Response: Thanks for the valuable suggestion and we have added it as following: Each treatment group contained 10 seedlings, and four independent biological replicates were performed.

  1. L342, delete “and obtained similar results”.

Response: Thanks for the valuable suggestion and we have deleted it as suggested.

  1. L351, 354, Please provide the brand, company and country of the main reagent.

Response: Thanks for the valuable suggestion and we have added it as suggested.

  1. L381, It should be 4.7.

Response: Thanks for the valuable suggestion and we have corrected it.

Reviewer 2 Report

The MS entitled "Coronatine enhances chilling tolerance of tomato plants by inducing chilling-related epigenetic adaptations and transcriptional reprogramming " is a nice piece of work done by the authors.

Cold stress is abiotic stress that plants experience during extended exposure to freezing temperatures. A frost is especially harmful in late spring or early fall when plants are in an active growth stage. Exposure to cold stress-induced signalling mechanisms through an increase in cytosolic Ca2+. The increase in cytosolic Ca2+ can be triggered through the activation of cyclic nucleotide-gated ion channels (CNGCs) in response to cold stress

In my opinion, the overall concept is interesting and important. The paper is well written and is a worthy contribution that will be of interest to the readers of IJMS journal.

The paper requires only Minor revisions to be acceptable for publication, I have a few minor suggestions which I believe will improve the manuscript.
The abstract can serve as a stand-alone document that succinctly describes both procedures and conclusions. Highlights are specific and present new insights. In general, the paper is clear and well-written.

My suggestion regarding this MS is to accept after revising with the following points:

1. Authors presented a very good introduction, but they could explain more about molecular mechanisms of plant temperature tolerance and potential mitigation from chilling/frost toxicity.

2. Introduction and discussion sections authors could involve also new aspects - hormonal regulation of the arsenic tolerance (interaction with SA, CYT etc), including new references.

3. Add more view on photosynthetic parameters/discuss, which kind of parameters are more sensitive and why.

4. Add some recent references in the MS. It is important to discuss plant regulation mechanisms. The paper brings many new aspects and the novelty of the paper is OK, but I would like to invite the authors to discuss also more eco-physiological aspects using new references: DOI: 10.1093/plphys/kiab532; 10.1080/15592324.2020.1807722; 10.1007/s00344-021-10319-0; 10.1038/s41598-020-63006-7; 10.3390/ijms20102479

5. I recommend enlarging and splitting some pictures - de facto illegible.

6. Have you tried the HSD Tukey test? It would probably be more significant than the student test.

7. Fig 1a - missing scale and the pictures are duplicated in the text... which is confusing.

Final COMMENTS
- The manuscript is useful and innovative, it contains original data.
- This study presents the relevant matter in more depth than some of the other related publications, therefore, I recommend its publication after MODERATE  changes.

Author Response

Reviewer 2

The MS entitled "Coronatine enhances chilling tolerance of tomato plants by inducing chilling-related epigenetic adaptations and transcriptional reprogramming " is a nice piece of work done by the authors.

Cold stress is abiotic stress that plants experience during extended exposure to freezing temperatures. A frost is especially harmful in late spring or early fall when plants are in an active growth stage. Exposure to cold stress-induced signalling mechanisms through an increase in cytosolic Ca2+. The increase in cytosolic Ca2+ can be triggered through the activation of cyclic nucleotide-gated ion channels (CNGCs) in response to cold stress

In my opinion, the overall concept is interesting and important. The paper is well written and is a worthy contribution that will be of interest to the readers of IJMS journal.

The paper requires only Minor revisions to be acceptable for publication, I have a few minor suggestions which I believe will improve the manuscript.

The abstract can serve as a stand-alone document that succinctly describes both procedures and conclusions. Highlights are specific and present new insights. In general, the paper is clear and well-written.

My suggestion regarding this MS is to accept after revising with the following points:

  1. Authors presented a very good introduction, but they could explain more about molecular mechanisms of plant temperature tolerance and potential mitigation from chilling/frost toxicity.

Response: Thanks for the valuable suggestion and we have added more relative reference as suggested:

When exposure to low temperature, plants synthesize a large number of protective substances to enhance cold tolerance. The increase of soluble sugars such as sucrose, raffinose, and trehalose, enhances plant cold tolerance by maintaining cell osmotic potential and cell membrane stability (Kaplan and Guy, 2004; Kaplan et al., 2007). Exogenously application of oligosaccharides including chitosan oligosaccharide (CSOS), cello-oligosaccharide (COS), and xylooligosaccharide (XOS) effectively increases the content of soluble sugar and plant chilling tolerance (He et al., 2021). Under chilling stress, plants accumulate polyamines to increase resistance to chilling (Gao et al., 2020; Sheteiwy et al., 2017). In addition, plants enhance the activity of antioxidant enzymes to prevent cold-induced oxidative stress and photosynthetic damage (Hajihashemi et al., 2020; Kaniuga, 2008).

  1. Introduction and discussion sections authors could involve also new aspects - hormonal regulation of the arsenic tolerance (interaction with SA, CYT etc), including new references.

Response: Sorry, hormonal regulation of the arsenic tolerance is a very interesting research direction, and we may be involved in this research in the future. However, this manuscript is focus on the mechanism of plants chilling tolerance, it does not involve arsenic poisoning, and we have not obtained any research data related to arsenic poisoning, so we cannot effectively discuss arsenic poisoning based on our data. At the same time, this manuscript has cited 84 literatures. We are afraid that references that deviated from the research content of this manuscript will distract readers, so we have not added references and discussion related to arsenic poisoning in this manuscript.

  1. Add more view on photosynthetic parameters/discuss, which kind of parameters are more sensitive and why.

Response: Thanks for the valuable suggestion and we have added more relative reference as suggested: Plant photosynthesis is sensitive to low temperature (Ensminger et al., 2006). The photosynthetic parameters including maximum quantum yield of photosystem II (Fv/Fm), performance indices (PIABS), total chlorophylls (Chls), net photosynthesis (PN), intercellular CO2 concentration (Ci), and water use efficiency (WUE) are significantly reduced by cold stress (Hajihashemi et al., 2020; Hajihashemi et al., 2018). The decrease of these photosynthetic parameters is recovered under long term cold treatment (Hajihashemi et al., 2020), which maybe relate to cold acclimation. Some of the COR genes encode chloroplast associated proteins such as COR15a and COR15b, that protect chloroplast membrane and stromal proteins, thus reducing the inhibition of photosynthesis caused by cold stress (Thalhammer et al., 2014; Thomashow et al., 1996). In this study, coronatine-induced chilling responsive genes also contained chloroplast associated genes, including CYTOCHROME B5 REDUCTASE, LIPOXYGENASE 3, and ATP CARRIER PROTEIN 1, which may function in mitigating the damage to the photosystem and the inhibition of photosynthesis under chilling stress.

Since these photosynthesis parameters are directly coupled with photosynthesis, the sensitivity of these parameters has not shown much different in previous studies (Hajihashemi et al., 2020; Hajihashemi et al., 2018). Because the sensitivity of the photosynthetic parameters is not the subject of this manuscript, we just discussed it in the response without putting it in the manuscript to avoid getting off topic.

  1. Add some recent references in the MS. It is important to discuss plant regulation mechanisms. The paper brings many new aspects and the novelty of the paper is OK, but I would like to invite the authors to discuss also more eco-physiological aspects using new references: DOI: 10.1093/plphys/kiab532; 10.1080/15592324.2020.1807722; 10.1007/s00344-021-10319-0; 10.1038/s41598-020-63006-7; 10.3390/ijms20102479

Response: Thanks for the valuable suggestion and we have added more relative reference as suggested:

When exposure to low temperature, plants synthesize a large number of protective substances to enhance cold tolerance. The increase of soluble sugars such as sucrose, raffinose, and trehalose, enhances plant cold tolerance by maintaining cell osmotic potential and cell membrane stability (Kaplan and Guy, 2004; Kaplan et al., 2007). Exogenous application of oligosaccharides including chitosan oligosaccharide (CSOS), cello-oligosaccharide (COS), and xylooligosaccharide (XOS) effectively increases the content of soluble sugar and plant chilling tolerance (He et al., 2021). Under chilling stress, plants accumulate polyamines to increase resistance to chilling (Gao et al., 2020; Sheteiwy et al., 2017). In addition, plants enhance the activity of antioxidant enzymes to prevent cold-induced oxidative stress and photosynthetic damage (Hajihashemi et al., 2020; Kaniuga, 2008).

JA reduces chilling injury of several tropical and subtropical fruits, such as mango, banana, and tomato fruits (Ding et al., 2002; Gonzalez-Aguilar et al., 2000; Ruan et al., 2019; Zhao et al., 2013). MeJA treatment significantly induce the expression of many cold-responsive genes under cold storage to reduce the damage of plants caused by cold (Ruan et al., 2019).

  1. I recommend enlarging and splitting some pictures - de facto illegible.

Response: Thanks for the valuable suggestion and we have split previous Figure 5 and Figure 7 and enlarged Figure 4-9 as suggested.

  1. Have you tried the HSD Tukey test? It would probably be more significant than the student test.

Response: Thanks for the valuable suggestion and we have used HSD Tukey test instead of student test as suggested.

  1. Fig 1a - missing scale and the pictures are duplicated in the text... which is confusing.

Response: Thanks for the valuable suggestion and we have added the scales as suggested. In Fig 1a, the picture above shows the representative tomato seedlings (selected from 10 seedlings per treatment) before exposure to 4°C, and the picture below shows the representative tomato seedlings (selected from 10 seedlings per treatment) after exposure to 4°C. The picture above is to show the growth of tomato seedlings before 4°C treatment, indicating that these seedlings did not have significant growth differences before 4°C treatment. The six tomato seedlings shown in one picture were randomly selected from each of the six groups respectively and photographed at the same time, with no duplication between them. But some of the tomato seedlings between pictures above and below may be duplicates because they were randomly selected before and after 4°C treatment. We have added relevant explanations in the figure legends.

Final COMMENTS

- The manuscript is useful and innovative, it contains original data.

- This study presents the relevant matter in more depth than some of the other related publications, therefore, I recommend its publication after MODERATE changes.

Response: Thanks for your recognition of our manuscript.

Reference:

Ding, C.K., Wang, C.Y., Gross, K.C., and Smith, D.L. (2002). Jasmonate and salicylate induce the expression of pathogenesis-related-protein genes and increase resistance to chilling injury in tomato fruit. Planta 214, 895-901.

Du, M., Zhao, J., Tzeng, D.T.W., Liu, Y., Deng, L., Yang, T., Zhai, Q., Wu, F., Huang, Z., Zhou, M., et al. (2017). MYC2 orchestrates a hierarchical transcriptional cascade that regulates jasmonate-mediated plant immunity in tomato. Plant Cell 29, 1883-1906.

Ensminger, I., Busch, F., and Huner, N.P.A. (2006). Photostasis and cold acclimation: sensing low temperature through photosynthesis. Physiol Plant 126, 28-44.

Gao, C., Sheteiwy, M.S., Han, J., Dong, Z., Pan, R., Guan, Y., Alhaj Hamoud, Y., and Hu, J. (2020). Polyamine biosynthetic pathways and their relation with the cold tolerance of maize (Zea mays L.) seedlings. Plant Signal Behav 15, 1807722.

Gonzalez-Aguilar, G.A., Fortiz, J., Cruz, R., Baez, R., and Wang, C.Y. (2000). Methyl jasmonate reduces chilling injury and maintains postharvest quality of mango fruit. J Agric Food Chem 48, 515-519.

Hajihashemi, S., Brestic, M., Landi, M., and Skalicky, M. (2020). Resistance of Fritillaria imperialis to freezing stress through gene expression, osmotic adjustment and antioxidants. Sci Rep 10, 10427.

Hajihashemi, S., Noedoost, F., Geuns, J.M.C., Djalovic, I., and Siddique, K.H.M. (2018). Effect of cold stress on photosynthetic traits, carbohydrates, morphology, and anatomy in nine cultivars of Stevia rebaudiana. Front Plant Sci 9, 1430.

He, J., Han, W., Wang, J., Qian, Y., Saito, M., Bai, W., Song, J., and Lv, G. (2021). Functions of oligosaccharides in improving tomato seeding growth and chilling resistance. J Plant Growth Regul 41, 535-545.

Kaniuga, Z. (2008). Chilling response of plants: importance of galactolipase, free fatty acids and free radicals. Plant Biol (Stuttg) 10, 171-184.

Kaplan, F., and Guy, C.L. (2004). beta-Amylase induction and the protective role of maltose during temperature shock. Plant Physiol 135, 1674-1684.

Kaplan, F., Kopka, J., Sung, D.Y., Zhao, W., Popp, M., Porat, R., and Guy, C.L. (2007). Transcript and metabolite profiling during cold acclimation of Arabidopsis reveals an intricate relationship of cold-regulated gene expression with modifications in metabolite content. Plant J 50, 967-981.

Ruan, J., Zhou, Y., Zhou, M., Yan, J., Khurshid, M., Weng, W., Cheng, J., and Zhang, K. (2019). Jasmonic acid signaling pathway in plants. Int J Mol Sci 20.

Sheteiwy, M., Shen, H.Q., Xu, J.G., Guan, Y.J., Song, W.J., and Hu, J. (2017). Seed polyamines metabolism induced by seed priming with spermidine and 5-aminolevulinic acid for chilling tolerance improvement in rice (Oryza sativa L.) seedlings. Environ Exp Bot 137, 58-72.

Song, Z., Lai, X., Yao, Y., Qin, J., Ding, X., Zheng, Q., Pang, X., Chen, W., Li, X., and Zhu, X. (2022). F-box protein EBF1 and transcription factor ABI5-like regulate banana fruit chilling-induced ripening disorder. Plant Physiol 188, 1312-1334.

Thalhammer, A., Bryant, G., Sulpice, R., and Hincha, D.K. (2014). Disordered cold regulated15 proteins protect chloroplast membranes during freezing through binding and folding, but do not stabilize chloroplast enzymes in vivo. Plant Physiol 166, 190-201.

Thomashow, M.F., Artus, N., Gilmour, S., Stockinger, E., Wilhelm, K., Zarka, D., Joseph, R.A., Uemura, M., and Steponkus, P.L. (1996). Function and regulation of Arabidopsis thaliana COR (cold-regulated) genes. Plant Physiol 111, 81003-81003.

Zhang, X., Fowler, S.G., Cheng, H., Lou, Y., Rhee, S.Y., Stockinger, E.J., and Thomashow, M.F. (2004). Freezing-sensitive tomato has a functional CBF cold response pathway, but a CBF regulon that differs from that of freezing-tolerant Arabidopsis. Plant J 39, 905-919.

Zhao, M.L., Wang, J.N., Shan, W., Fan, J.G., Kuang, J.F., Wu, K.Q., Li, X.P., Chen, W.X., He, F.Y., Chen, J.Y., et al. (2013). Induction of jasmonate signalling regulators MaMYC2s and their physical interactions with MaICE1 in methyl jasmonate-induced chilling tolerance in banana fruit. Plant Cell Environ 36, 30-51.

Round 2

Reviewer 1 Report

This version can be accepted for publication.